# Evolutionary Features for Dynamic Link Prediction in Social Networks

Nazim Choudhury [1] 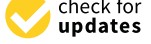 and Shahadat Uddin [2,*] 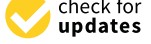

1  Department of Computer Science, University of Wisconsin-Green Bay, Green Bay, WI 54302, USA
2  Faculty of Engineering, School of Project Management, University of Sydney, Sydney 2037, Australia
*  Correspondence: shahadat.uddin@sydney.edu.au

**Abstract:** One of the inherent characteristics of dynamic networks is the evolutionary nature of their constituents (i.e., actors and links). As a time-evolving model, the link prediction mechanism in dynamic networks can successfully capture the underlying growth mechanisms of social networks. Mining the temporal patterns of dynamic networks has led researchers to utilise dynamic information for dynamic link prediction. Despite several methodological improvements in dynamic link prediction, temporal variations of actor-level network structure and neighbourhood information have drawn little attention from the network science community. Evolutionary aspects of network positional changes and associated neighbourhoods, attributed to non-connected actor pairs, may suitably be used for predicting the possibility of their future associations. In this study, we attempted to build dynamic similarity metrics by considering temporal similarity and correlation between different actor-level evolutionary information of non-connected actor pairs. These metrics then worked as dynamic features in the supervised link prediction model, and performances were compared against static similarity metrics (e.g., AdamicAdar). Improved performance is achieved by the metrics considered in this study, representing them as prospective candidates for dynamic link prediction tasks and to help understand the underlying evolutionary mechanism.

**Keywords:** evolutionary features; dynamic network; dynamic link prediction; social network; actor dynamicity

## 1. Introduction

Systems, regardless of being physical or abstract, in many real-world domains, including sociology, biology, criminology, informatics, economics and communication, can be mapped into a network. In these networks, nodes represent the individuals or actors, and links or edges represent various types of relations or interactions among them. These systems are inherently dynamic, where evolution occurs due to concurrent arrivals and/or removals of actors and simultaneously forming, strengthening, weakening, and dissolving ties among network actors over time. Network science caters to various methods supporting the study and modeling of a network evolution process that governs their dynamics [1]. Among them, link prediction is the fundamental computational problem that models the underlying growth mechanism of evolving networks [2]. The primary objective of link prediction methods is to estimate the likelihood of the emergence of new links among network actors utilizing the observed links, actor attributes, network structure, topology, or nodal properties [3]. This likelihood is measured in regard to similarity or proximity between non-connected node pairs, predominantly calculated using topological properties or probabilistic models [4]. The link prediction mechanism in social networks has gained considerable interest not only for mining and analyzing the network evolution in particular but also due to its wide variety of applications. These include recommendation systems, anomaly detection, influence analysis, and community detection [5], predicting linkage patterns in scientific collaboration networks [6], social security networks [7], disease spreading networks, especially the COVID-19 contact network [8], identifying hidden groups in

terrorist or criminal networks [9], discovering new protein interactions [10], understanding connectome patterns (mapping of neural connections in the brain) of organism's nervous system [11], improving transportation efficiency by efficient routing strategy [12], predicting users' ad-clicking actions and recommending interesting web contents for marketing purpose, and in sensor networks, ensuing information transfer secrecy [13] and optimal routing [14]. Subsequently, network science communities have proposed a wide range of similarity metrics and prediction strategies [15]. However, a vast magnitude of them focuses on static networks. There are two major hindrances to these prediction methods. Firstly, they depend on feature engineering over actors' network and non-network-based attributes by most of the classification and regression methods utilized for supervised and unsupervised link prediction [16]. Secondly, they do not acknowledge the dynamicity of a network resulting from changes in its past behaviours over time [17].

Although link prediction is a time-evolving network analysis model that measures the probability of future links by analyzing the existing links in the network, traditional similarity metrics-based methods generally overlook taking the evolutionary information of dynamic networks into account. Dynamic networks evolve through simultaneous arrivals and/or departure of actors as well as the creation and/or deletion of links among these actors. These time-varying networks, also known as "temporal" or "longitudinal" networks, are suitable for describing entities whose dynamics change over time. The combination of the technical possibility of storing, processing, and representing large-scale datasets and the increased proliferation and ubiquity of real-world dynamic network datasets has led dynamic network analysis to gain considerable research interest to understand the underlying mechanisms of their evolutionary dynamics better. Recently, researchers have attempted the issue of ***dynamic link prediction*** or link prediction in dynamic networks. Dynamic link prediction is the process of inferring the possibility of future links among the dynamic entities or network actors through exploring historical or temporal information [18]. Different dynamic link prediction methods explore a wide range of techniques. Most of the techniques used a wide variety of structural and network topological features to compute the likelihood of link formations. For example, Zhang et al. [19] used a node (i.e., actor) centrality-based temporal link prediction where the authors distinguished the contributions of common neighbors to connection likelihood by their eigenvector centralities. By considering the importance of nodes as the probability of attracting other nodes, Wu et al. [20] used an eigenvector-based node ranking strategy along with a forecasting method called Adaptive Weighted Moving (AWM) for dynamic link prediction. Chi et al. [21] categorized the nodes into different levels based on the influence strength of the node compared to its neighbors that change over time. The authors computed the connection probability between a pair of nodes using their corresponding levels of influence strength and the attraction force between them to predict the missing links in dynamic networks. Chen and Li [22] formulated the link prediction problem in dynamic networks as an optimization problem that not only collectively leveraged the structural and temporal information to better infer a low-rank representation for each node but also preserved the deep network structure via high-order proximity among nodes. The authors also used an efficient block coordinate gradient descent approach to address the optimization problem.

Among other methods, researchers exploited the collective influence, the community walk features, and the centrality features [23], subgraph evolution [24], effective influence mechanism in relation to actors' degree and strong connectivity across short and long path among them [25], 'graphlet' transitions [26], dynamic latent space representation of actors and random walk in temporal networks [27], the correlation between different types of links along with temporal features (e.g., "recency", temporal activeness) [28], probabilistic temporal measure [29], probabilistic generative model [30], matrix and tensor factorization [31] and deep learning techniques [32].

Nevertheless, some of them are subject to their inherent limitation. For example, probabilistic models involve the prior definition of link occurrences' distribution, which is problematic for temporal networks. Furthermore, the probabilistic model is only suitable

for small networks with a few hundred actors. Similarly, matrix or tensor-based methods are not feasible for real-time link predictions in large networks due to the computational complexity and time requirements [33]. Supervised link prediction techniques in dynamic networks [34,35] take advantage of either temporal sequences or temporal variations of several network topological properties (e.g., Commonneighbors, Jaccard, AdamicAdar), used to measure similarity/proximity between actor pairs in static networks, to train the classifier instead of measuring their similarity/proximity by mining actor-level evolutionary aspects including temporal patterns of neighbourhood changes or evolutionary community-aware information. Some of these techniques [36,37] included a time series forecasting method to predict the future values of topological changes to training classifiers for supervised link prediction. This exercise can be counterproductive since the prediction is performed using predicted and unrealistic values. To address these issues, this study attempted to define dynamic features by employing temporal similarity and correlation of actor-level evolution (e.g., temporal structural and neighbourhood changes) experienced by individual actors in dynamic networks.

Actors in dynamic networks are subject to varying temporal changes (i.e., dynamicity) within the temporal network snapshots due to variations of different network activities (e.g., link formation, link deletion) over time. This leads to temporal changes in actors' positions and neighbourhoods in dynamic networks. Therefore, actor-level dynamicity triggers both micro (e.g., neighbourhood changes) and mesoscopic changes (e.g., community participation) in dynamic networks. By mining the similarity or correlation between these diverse actor-level temporal fluctuations (i.e., structural position and neighbourhood), it is possible to generate dynamic similarity metrics, similar to the topological similarity metrics computed in traditional link prediction in static networks, for the purpose of dynamic link prediction in dynamic networks. This study first analyses dynamic networks to develop such metrics by considering the evolutionary information attributed to actor pairs to define two different types of actor-level dynamicity. It then identifies the similarity and correlation between the temporal sequences of dynamicity values. These metrics denote the similarity and/or proximity between actor pairs regarding their evolving features within temporal networks. The research question this study attempts to address is whether the likelihood of future links among non-connected actor pairs in dynamic networks depends on the similarity or correlation of their evolving features regarding actor-level network structure and neighbourhood. Dynamic similarity metrics built in this study were then applied to both directed and undirected dynamic networks. A supervised link prediction framework was set out to successfully predict future links among non-connected actor pairs in dynamic networks. Performances of dynamic metrics constructed in this study were compared against the well-known traditional static similarity metric (i.e., AdamicAdar), including the time series-based link prediction method.

## 2. Dynamic Similarity Metrics

In the case of link prediction in static networks, there are predominantly three types of topological similarity indices [3]; (i) local, (ii) global, and (iii) quasi-local. Local similarity indices are constructed using neighbourhood-related structural information, whereas global similarity indices use the whole network topological information to compute the similarity between actor pairs. Notwithstanding, many real-world networks are longitudinal in nature, involve dynamic processes, and evolve temporally. With this temporal evolution of networks, actors simultaneously experience altering topological properties that make these similarity indices incompetent in dynamic link prediction. Researchers have used time series of these topological similarity indices to emulate the evolutionary aspect of dynamic networks. A given dynamic network can, therefore, be defined as a time series of network snapshots where each snapshot represents the corresponding network state at a particular time, known as a short interval network (SIN). Actors change their link structure, neighbourhood, and network positions in every SIN over time that, in turn, also represent 'actor dynamicity' [38]. The term actor dynamicity refers to the variability in

the involvement of individual actors in dynamic social networks over time. Based on the temporal patterns and evolutionary processes taking place in dynamic networks, firstly, this study attempts to define two types of actor-dynamicity, namely (i) structural dynamicity, and (ii) neighbourhood dynamicity. Secondly, temporal similarity and correlation measures over this actor-level dynamicity are used to measure the similarity and/or proximity between non-connected actor pairs. In the following sections, different techniques are described which were used to build dynamic similarity metrics in this study.

### 2.1. Actor-Level Evolution in a Dynamic Network

As mentioned earlier, a dynamic network can be defined as a temporal sequence of small-scale network snapshots where each snapshot is known as a short-interval network (SIN). Actors experience variances in their link structure, altering neighbourhoods and subsequently changing network positions over time. The increasing size and complexity of modern dynamic networks have instigated the mechanism of splitting a large network into small-scale manageable components that facilitate the visualization and inference procedure. It simplifies the expedition of different aspects of the network to describe it succinctly without computational difficulties. Modifications of actors' network positions in SINs over time due to the varying nature of performing network activities (i.e., link formation, link deletion) and changing neighbourhoods are visualized in Figure 1. The top row in this figure represents a time series of three network snapshots $G_1$, $G_2$, $G_3$, known as short-interval networks (SIN), where these evolutionary networks are analysed to predict a future link between actor $a_1$ and $a_2$ at timestamp $t = 4$ in $G_4$. The bottom row represents the aggregated networks at timestamps $t = 2, 3$ where the first network denotes a union of $G_1$, $G_2$ (i.e., $G_1 \cup G_2$), and the second one denotes a union of $G_2$, $G_3$ (i.e., $G_2 \cup G_3$). Each network snapshot is accompanied by three centrality measures (i.e., degree, closeness, and betweenness) and neighbourhood incident to actor $a_1$ and $a_2$ at different timestamps both in individual short-interval networks and in aggregated networks. In the top row of this figure, the link structure and neighbourhoods of two actors (i.e., $a_1$ and $a_2$) are portrayed in three different SINs at three different timestamps (i.e., $t_1$, $t_2$ and $t_3$). The numbers on top of each SIN segment denote three different centrality measures (i.e., degree, closeness, and betweenness) for these two actors, and at the bottom of these three SINs, the neighbouring actors of these two actors are presented. Three centrality measures were computed using the networkX [39] package that supports the exploration and analysis of networks and network algorithms. For example, according to networkX, the degree centrality of an actor is the fraction of nodes it is connected to, and the closeness of an actor is the reciprocal of the average shortest path distance to that actor over other $n - 1$ reachable nodes by considering $n$ number of actors in the network. It is observable that the varying network positions of actors in temporal networks can be mapped by the centrality measures effectively. For example, actor $a_2$ experienced a higher degree and closeness centrality in comparison to $a_1$ in SIN $G_1$, whereas actor $a_1$ achieved a higher measure value in SIN $G_2$ in comparison to $a_2$. Conversely, although both achieved a similar degree centrality in SIN $G_3$, their closeness and betweenness centrality measures vary according to their network positions. Likewise, actor $a_1$ gained a new neighbourhood in $G_2$ which is different from what it had in $G_1$. Furthermore, in $G_3$, it regained one of its previous neighbourhoods (i.e., $a_6$). Simultaneously, despite its retention of previously gained neighbors (i.e., $a_6$) in $G_3$, it lost one of its neighbors from $G_2$ (i.e., $a_8$). Similar observations are evident in the case of actor $a_2$. Therefore, it is believed that actors experience variable changes in their network positions and neighbourhoods due to micro-level network activities. Incorporating the aforementioned observations, evident from Figure 1, and considering the network structural changes and altering neighbourhood over time in a series of network snapshots, this study attempts to define two types of actor-level dynamicity by mining the evolutionary information incident to individual actors in a dynamic network. These are namely (i) structural and (ii) neighbourhood dynamicity. In the following sections, this study describes the dynamicity measures.

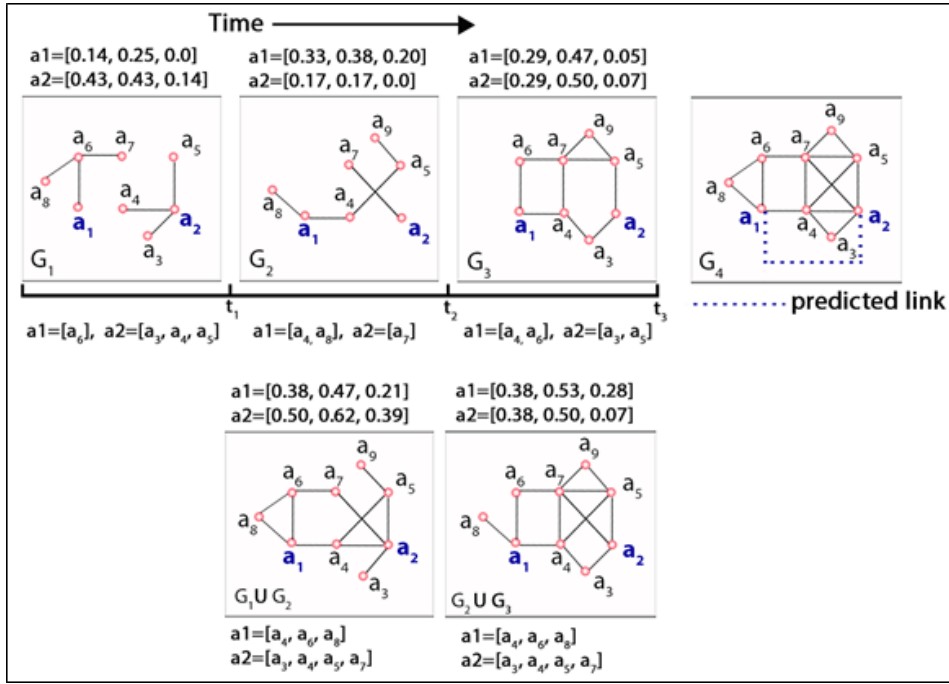

**Figure 1.** Visualisation of a dynamic link prediction framework considering a series of evolutionary network snapshots at different discrete timestamps ($t = 1, 2, 3, 4$) using abstract data.

### 2.1.1. Structural Dynamicity

The change in link structure and network positions experienced by actors in every SIN over time can be measured using different network measures utilized in social network analysis [38]. Therefore, this study used the following composite measure to quantify an actor's structural position in a network snapshot:

$$C_t^a = \begin{cases} deg_a(G_t) + cls_a(G_t) + btwn_a(G_t) & a \in V_t \\ 0 & a \notin V_t \end{cases} \tag{1}$$

where $C_t^a$ denotes network status/position of actor $a$ in a SIN, $G_t(V_t, E_t)$ at time $t$; $deg_a(G_t)$, $cls_a(G_t)$, and $btwn_a(G_t)$ denote the degree, closeness and betweenness centrality of actor $a$ in $G_t$. The underlying reasons for using a composite measure as a combination of three centrality measures are, firstly, that these measures are well-defined and can successfully quantify an actor's connectivity, position, communication dynamics, influence and broadcasting capabilities, and importance in a network, and, secondly, that these measures are correlated. For example, an actor with high betweenness and low closeness centrality can monopolize links from a small number of actors to many others. Likewise, a high degree with low closeness centrality denotes that the actor is embedded in a cluster far from the rest of the network. It is noteworthy that, in the case of a directed network, $deg_a(G_t)$ can be either in-degree or out-degree centrality measures of actors, or it can be a combination of both. This study considered the aggregation of in-degree and out-degree centrality measures together.

Motivated by the perception of these actor-oriented dynamic changes, the concept of an actor's positional dynamicity was developed by Uddin et al. [38] to quantify the temporal variations considering both dynamic and static social network topology. The underlying reason is that, according to social network topology, a dynamic network needs to be analysed in regard to the temporal aggregation of links among its actors [40], and simultaneously, different aspects of the dynamicity of dynamic networks can be quantified using both static and dynamic topology of social network analysis [41]. Furthermore, Chen et al. [42] used local topological similarity indices (e.g., AdamicAdar, Jaccard Coefficient), and unlike other supervised dynamic link prediction methods ([34,36]), instead of building time series of these indices, they considered their variations between adjacent

time steps. To find out the intrinsic relationship between the structural variations and the formation of links between non-connected actor pairs, the authors also defined a measure to quantify the rate of change of the structural properties. Following that, this study defined structural dynamicity as the rate or degree of actor-level structural changes computed at time $t$ using the following equation:

$$\delta^a(t) = \frac{C_t^a - C_{t-1}^a}{C_{t\cup t-1}^a} \qquad (2)$$

where $\delta^a(t)$ denotes the rate of structural dynamicity demonstrated by an actor $a$ at timestamp $t$. $C_t^a$ denotes the composite centrality measure, defined in Equation (1), incident to actor $a$ in a network snapshot $G_t$ at time $t$, whereas $C_{t-1}^a$ denotes the same centrality measure incident to actor $a$ in a network snapshot at time $t-1$ (i.e., previous timestamp) and finally, $C_{t\cup t-1}^a$ denotes the same measure of actor $a$ in an aggregated network $G_t \cup G_{t-1}$. This basically quantifies the structural and/or topological changes of every actor that it experiences at every timestamp in temporal networks. It calculates the rate of changes in an actor's topological importance measured via three centralities at a timestamp as the ratio between the difference in centralities in consecutive SINs and the centralities measured in the aggregated network over these SINs. The denominator in the structural dynamicity normalizes the rate of topological changes of an actor over time. Interested readers can refer to [38] for further explanation.

In Figure 1, the composite measure, defined in Equation (1), of actor $a_1$ is 0.39 in $G_1$, whereas in $G_2$, it is 0.91 for the same actor. Similarly, from the bottom row of this figure, we obtain the composite measure of this actor in the aggregated network (i.e., $G_1 \cup G_2$) at timestamp $t = 2$ as 1.06. Therefore, at time $t = 2$, the degree of structural evolution for actor a1 can be measured as $(\frac{|0.91-0.39|}{1.06}) = 0.491$. Similarly, at timestamp $t = 3$, the degree of structural evolution experienced by the same actor is measured as $(\frac{|0.81-0.91|}{/}1.19 = 0.084)$. Evidently, from the figure, actor a1 demonstrates a greater rate of dynamicity at the timestamp $t_2$ rather than $t_3$. Thus, the numerator of Equation (2) quantifies the rate of positional changes of an actor over adjacent SINs, and the denominator normalizes the change using the composite centrality values of that actor that it was supposed to acquire in an aggregated network without considering the diminution.

### 2.1.2. Neighbourhood Dynamicity

In social network analysis, the neighbourhood is defined as the local regions around individual actors considering different path lengths [43]. The neighbourhood also includes all the links among all the actors having a direct connection with egos. Neighborhood-based analysis within SINs can disclose different aspects of networks, including interesting features (e.g., local leadership changes, spurious/irregular activities) and structures not available from the aggregated global network [44]. In this study, we considered the neighbourhood as an individual's immediate field of interaction (i.e., at a distance). Subsequently, the neighbourhood dynamicity of an actor is measured in a SIN at timestamp ($t > 1$) as the ratio of an actor's total neighbor count in $G_t$ in comparison to the total neighbor count in an aggregated network at timestamp $t$. This ratio is further quantified with the neighbourhood gaining rate at timestamp $t$ in regard to the total number of actors in $G_t$. Thus, the neighbourhood dynamicity $\lambda^a(t)$ of actor $a$ at timestamp $t$ is defined as:

$$\lambda^a(t) = \frac{|N^a(G_t)|}{|N^a(G_1 \cup G_2 \cup G_3 \cup \ldots G_t)|} \times \frac{1}{V_t - N^a(G_t)} \qquad (3)$$

where $N^a(G_t)$ denotes the set of neighbors of actor $a$ (i.e., the neighbourhood of $a$), $G_t$ denotes a SIN $G$, and $V_t$ denotes the total number of actors in the SIN at timestamp $t$. The denominator in the first part of the Equation (3) denotes the neighbourhood of actor $a$ in an aggregated network comprised of all network snapshots before and at timestamp $t$ (i.e., $G_1 \cup G_2 \cup G_3 \ldots \cup G_t$). Similar to structural dynamicity, in the case of directed networks,

this study considered both in-degree and out-degree neighbors together. From Equation (3), we can observe that an actor can have a maximum score of one as neighbourhood dynamicity if it forms an association with every other actor in SIN at timestamp $t = 1$, maintains its neighbourhood in all the subsequent SINs in the dynamic network, and simultaneously form an association with every new actor appearing in the subsequent SINs. On the other hand, if an actor does not participate in any SIN, its neighbourhood dynamicity score will be counted as zero. From this equation, it is apparent that associations with more new actors in SINs and maintaining the already gained neighbourhood in subsequent SINs will boost the actor's neighbourhood dynamicity score. It is noteworthy that, for the first SIN, the aggregated network in the denominator of Equation (3) consists of only one and the first network snapshot. Therefore, for the first SIN where an actor crops up, the first part of this equation before the multiplication sign assigns a value of one for that actor. For example, considering Figure 1, in the case of actor $a_1$, the neighbourhood dynamicity at $t = 1$ is computed as ($\frac{1}{1} \times \frac{1}{8-1} = 0.143$). Similarly, for $t = 2$ and 3, the neighbourhood dynamicity values for $a_1$ are 0.133 and 0.095. On the other hand, for $a_2$, the time series of neighbourhood dynamicity is $[0.2, 0.042, 0.083]$. Conversely, considering actor $a_9$, the temporal sequence of neighbourhood dynamicity is $[0, 0.167, 0.167]$, where this actor appears in $G_2$ for the first time and, therefore, for $G_1$, the dynamicity value is zero (0). As mentioned earlier, contrary to neighbourhood dynamicity, every actor is assigned a zero value for the structural dynamicity in the first SIN, irrespective of their appearance in that SIN.

### 2.2. Dynamic Features

In this section, we describe methods to define dynamic features for the purpose of link prediction by considering the evolutionary aspects of non-connected actor pairs defined in the previous section. These features will denote the similarity of proximity between actors in regard to their structural and neighbourhood evolution. To define the similarity and/or proximity between actor pairs, we compare the time series information comprised of both structural and neighbourhood dynamicity, as calculated above, incident to actor pairs. For example, according to Figure 1, to predict a link between actors $a_1$ and $a_2$ in $G_4$, this study builds two separate temporal sequences of $\delta^a(t)$ and $\lambda^a(t)$ (i.e., actor-level structural and neighbourhood dynamicity) incident to actor $a$ and calculated using Equations (2) and (3) over time. For these two actors, the temporal sequences of structural dynamicity are $a_1 = [1, 0.491, 0.091]$ and $a_2 = [1, 0.437, 0.436]$. It is noteworthy that, for the first timestamp, the structural dynamicity is assigned to one since no variation can be computed using the first SIN. The proximity between a pair of actors is defined in regard to the temporal similarity and correlation between two different temporal sequences encompassing their dynamicity values over time. In the following sections, three different methods are described that are used to compute the similarity/proximity between actor-level evolutionary information for non-connected actor pairs in regard to their aforementioned dynamicity values. Each method assigns a score $sim_i(a, b)$ to a pair of actors $(a, b)$ where the $i$th method computes the similarity or proximity between actors $a$ and $b$.

### 2.2.1. Temporal Similarity

In time series clustering, to measure the similarity between temporal sequences with varying speeds, existing distance measures (i.e., Euclidean, Manhattan) produce un-intuitive results and demonstrate incompetency in producing optimal alignment. For example, the Euclidean technique simply measures the distance between a pair of time series by summing the squared distances from each point in one-time series to the corresponding point in the other. If the pair are identical, with one being shifted along the time axis, Euclidean distance may consider totally different time sequences. The dynamic programming-based method of Dynamic Time Warping (DTW) overcomes the aforementioned limitation of traditional distance measures to provide intuitive distance measurements between temporal sequences by ignoring both global and local deviations in the time dimension [45]. It measures the similarity between two time series by shrinking

or expanding or simply "warping" the time axis of one (or both) sequences to achieve better alignment. This warping technique is an example of dynamic programming that can be used to determine corresponding regions between two time series to measure their similarity. Let $X^a$ and $Y^b$ be the time series of length $|m|$ and $|n|$ considering the chosen dynamicity measure, described earlier in Section 2.1 (i.e., structural and neighbourhood dynamicity), for actors $a$ and $b$, where $m, n \leq N$, and $N$ is the total number of SINs. If $d(x_i, y_j)$ denotes local distance measure (e.g., Euclidean, Minkowski), defined to compare two different points in $X^a$ and $Y^b$, then the goal of the DTW technique is to find an optimal alignment between $X^a$ and $Y^b$ with minimum overall distance. Details of this technique can be found in the study by Müller [46]. The notion of this alignment depends on the definition of an $(m, n)$-warping path which is a sequence $p = p_1, p_2, p_3, \ldots, p_l$ with $p_l = (m_l, n_l) \in [1:m]x[1:n]$ for $l \in [1:L]$, where $L$ denotes the length of warping path. The optimal warping path between $X^a$ and $Y^b$ is defined as a warping path $p^*$ with the minimum distance among all possible warping paths. To accomplish that, it may encounter that a single point in one time series may be mapped to multiple points of the other. The optimal warping path is determined by following a dynamic programming method that recursively measures the following function at every step:

$$\gamma(i, j) = d(x_i, y_i) + min[(\gamma(i-1, j-1), \gamma(i-1, j), \gamma(i, j-1))] \tag{4}$$

Here, $\gamma(i, j)$ represents the optimal warping path defined between the $i$th and $j$th component of two time series. In Figure 2, this study presents a comparable representation of calculating similarities between two temporal sequences using traditional distance measures (Figure 2a) (e.g., Euclidean) and the DTW method (Figure 2b). In this figure, the dashed lines represent the distance between corresponding points in both time series. The traditional approach aligns the $i$th point in one time series with the corresponding $j$th point of the other, whereas DTW provides nonlinear alignment to produce a more intuitive similarity measure and allows similar shapes to match ($i \rightarrow j, j + 1$) even if it requires localized stretching along the time axis. In DTW, the difference between these time series is the warped path distance which is measured by summing the distances between each pair of points connected by the dashed lines in the figure. Considering the temporal similarity measures using the DTW technique, this study first computes the structural and neighbourhood dynamicity of actor pairs at every timestamp of temporal network snapshots. Secondly, using temporal sequences of these dynamicity values, this study applied the DTW technique to measure temporal similarity between them. The temporal similarity between the time series of actors' dynamicity values will represent their evolutionary proximity or similarity. Therefore, values of first and second dynamic similarity metrics, developed in this study for actor pair $a$ and $b$ considering structural and neighbourhood dynamicity values, are defined as follows:

$$sim_1(a, b) = d_{p^*} \times (\delta_i^a, \delta_j^b) = min\left\{\sum_{l=1}^{L} d(\delta_{ml}^a, \delta_{nl}^b)\right\} \tag{5}$$

where $\delta_i^a$ and $\delta_j^b$ are $i$th and $j$th element of time series of structural dynamicity, and $m$ and $n$ denote the length of temporal sequences of structural dynamicity values incident to actor pairs $a$ and $b$, respectively. Similarly, the temporal similarity between neighbourhood dynamicity values over time represents this study's second dynamic similarity metric:

$$sim_2(a, b) = d_{p^*} \times (\lambda_i^a, \lambda_j^b) = min\left\{\sum_{l=1}^{L} d(\lambda_{ml}^a, \lambda_{nl}^b)\right\} \tag{6}$$

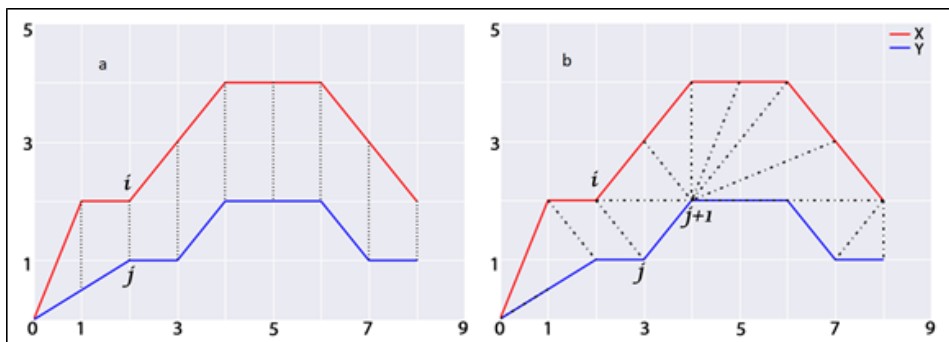

**Figure 2.** Visualizations of measuring similarity between two temporal sequences (**a**) traditional approach; (**b**) Dynamic Time Warping approach. Dashed lines represent the distance between corresponding points in both time series.

### 2.2.2. Correlation-Based Similarity

Correlation analysis is a statistical evaluation method that is used to quantify the strength and direction of the linear association between two variables. It is widely used in financial network analysis, asset allocation, portfolio optimisation and risk management [47]. This study applied correlation analysis to measure the affinities or similarities between actor pairs in regard to the temporal sequences of dynamicity values in all SIN. The assumption here is that two actors are similar if they fluctuate in a similar fashion considering any dynamicity measurement (i.e., dynamicity values of one actor increase or decrease with the other at the same time). If $\delta^a(t)$ and $\delta^b(t)$ denote the structural dynamicity, and $\lambda^a(t)$ and $\lambda^b(t)$ denote the temporal neighbourhood dynamicity of actor $a$ and $b$ at time $t$, then the evolution similarity between them is computed in regard to the Pearson correlation coefficient. Therefore, the third and fourth dynamic similarity metrics to measure the similarity of proximity between actor pairs $a$ and $b$ in this study are constructed as follows:

$$sim_3(a,b) = \frac{\sum_t [(\delta^a(t) - \overline{\delta^a})(\delta^b(t) - \overline{\delta^b})]}{\sqrt{\sum_t (\delta^a(t) - \overline{\delta^a})^2}\sqrt{\sum_t (\delta^b(t) - \overline{\delta^b})^2}} \qquad (7)$$

$$sim_4(a,b) = \frac{\sum_t [(\lambda^a(t) - \overline{\lambda^a})(\delta^b(t) - \overline{\lambda^b})]}{\sqrt{\sum_t (\lambda^a(t) - \overline{\lambda^a})^2}\sqrt{\sum_t (\lambda^b(t) - \overline{\lambda^b})^2}} \qquad (8)$$

Here, $\delta^a(t)$ and $\lambda^a(t)$ represent the structural and neighbourhood dynamicity, respectively, of actor $a$ at timestamp $t$. $\overline{\delta^a}$ represents the average structural dynamicity of actor $a$ over all SINs. Similarly, $\overline{\lambda^a}$ represents the average neighbourhood dynamicity of actor $a$ over all SINs.

### 2.2.3. Bray–Curtis Similarity

Although a significant amount of dynamic link prediction studies have exploited the static topological similarity metrics (e.g., CommonNeighbors, Jaccard Coefficient, Resource Allocation) over time to compute the similarity between actor pairs, this study uses an abundance-based similarity metric which is widely used in biology and ecology domain. Frequently used by marine ecologists to measure bio-diversity, the Bray–Curtis similarity measure was initially proposed by Bray and Curtis [48], which is principally employed in multivariate analysis of biological assemblage data and signifies the 'relativization' of species-wise differences in regard to their total abundance in biological metaphor [49]. However, using the Bray–Curtis method, the distance between two entities X and Y in regard to n-dimensional feature space can be determined as described by Legendre and Legendre in [50]:

$$BC_{XY} = \frac{\sum_{i=1}^{n}|x_i - y_i|}{\sum_{i=1}^{n}|x_i + y_i|} \tag{9}$$

where $x_i$ and $y_i$ denote the $i$th feature of $X$ and $Y$, respectively. The numerator signifies differences between X and Y in regard to the abundance of feature $i$, and the denominator normalizes the differences. In this study, we defined two dynamicity measures (i.e., structural and neighbourhood) as features. We consider these evolutionary aspects of actors in $T$ SINs to compute the similarity between them. In this study's context, the Bray–Curtis distance between actor $a$ and $b$ using two dynamicity measures is defined as:

$$BC_{ab} = \frac{\sum_{t=1}^{T}\left[\left|\delta^a(t) - \delta^b(t)\right| + \left|\lambda^a(t) - \lambda^b(t)\right|\right]}{\sum_{t=1}^{T}\left[\left|\delta^a(t) + \delta^b(t)\right| + \left|\lambda^a(t) + \lambda^b(t)\right|\right]} \tag{10}$$

Since the distance represents dissimilarity, $1 - BC_{ab}$ was used to represent similarity. Thus, the last dynamic similarity metric in this study is defined as follows:

$$sim_5(a, b) = 1 - \frac{\sum_{t=1}^{T}\left[\left|\delta^a(t) - \delta^b(t)\right| + \left|\lambda^a(t) - \lambda^b(t)\right|\right]}{\sum_{t=1}^{T}\left[\left|\delta^a(t) + \delta^b(t)\right| + \left|\lambda^a(t) + \lambda^b(t)\right|\right]} \tag{11}$$

## 3. Network Datasets and Experimental Settings

Considering two different time intervals $(t_1, t')$, $(t_t, t'_1)$ where $t_1 < t' < t'_1$ and a finite set of discrete time points within the range $[t_1 - t']$ as $T = t_1, (t_1 + \tau), (t_1 + 2\tau), \ldots,$ $(t_1 + n\tau), \ldots, (t' - \tau), t'$, where $\tau$ denotes the temporal sampling interval, a dynamic social network $G_T = (V, E_T)$ consists of a set of uniquely labeled actors $V = \{v_1, v_2, v_3, \ldots, v_n\}$, and $E_T = \{e_t(v_i, v_j, t) | v_i, v_j \in V; t \in T\}$, where $t$ represents the timestamp of a link $e$ between a pair of actors $(v_i, v_j)$. In addition, dynamic networks can be undirected, where $e = (v_i, v_j)$ and $e = (v_j, v_i)$ denote identical or directed links where these two links are not the same. Thus, the dynamic network is composed of an evolutionary sequence of network snapshots $G_T = \{G_{t_1}, G_{t_1+\tau}, G_{t_1+2\tau} \ldots G_{t_1+n\tau} \ldots G_{t'-\tau}, G_{t'}\}$ known as short-interval networks (SIN). Fluctuations in the total number of actors are taken into consideration across the time series of network snapshots. Any link may appear in multiple network snapshots at different timestamp(s). Considering this temporal sequence of network snapshots $G_T = \{G_{t_1}, G_{t_1+\tau}, G_{t_1+2\tau} \ldots G_{t_1+n\tau} \ldots G_{t'-\tau}, G_{t'}\}$, for a given pair of actors $(v_i, v_j)$, dynamic link prediction attempts to predict the link probability between them during the interval $(t', t'_1)$ in $G_{T+1}$ by analyzing the link formation and the temporal information in $G_T$ at timestamps $[t_1 - t']$ as $T = t_1, (t_1 + \tau), (t_1 + 2\tau), \ldots, (t_1 + n\tau), \ldots, (t' - \tau), t'$. Here, $G_T[t_1, t']$ and $G_T[t', t'_1]$ are considered as the networks in the training and testing phase, respectively. Traditional link prediction in the static network generally emphasizes the presence or absence of the links and simultaneously considers topological information to construe the similarity between actors. It does not consider the temporal information or the evolutionary dynamicity associated with all actors that vary in existence across network snapshots as well as in their associated links. A key aspect in dynamic link prediction is to generate dynamic similarity metrics (i.e., dynamic features) considering the evolutionary changes incident to actors. Therefore, this study attempts to develop such dynamic similarity metrics where the $i$th metric will assign a score $sim_i(v_i, v_j)$ to non-connected actor pairs $(v_i, v_j)$ considering the similarity/proximity of their evolutionary information in $G_T$. These scores will measure the likelihood of future links that emerge in $G_{T+1}$. In (Table 1), we summarized five dynamic similarity metrics/dynamic features constructed in this study to measure the similarity/proximity between non-connected actor pairs.

### 3.1. Datasets

For the dynamic network datasets collection, the 'KONECT Network Dataset' [51] (i.e., the Koblenz Network Collection) and 'Network Repository' [52] were used. KONECT project is run by the Institute of Web Science and Technologies at the University of Koblenz

as part of collecting large network datasets to facilitate research in network science and related fields. This study extracted different dynamic network datasets including directed and undirected networks where links between actors are timestamped.

**Table 1.** Five different values of $sim_i(a, b)$ computed by using five different dynamic similarity metrics. Each metric computes the similarity/proximity between non-connected actor pair $(a, b)$ by considering their structural (i.e., $\delta^a$, $\delta^b$) and neighbourhood (i.e., $\lambda^a$, $\lambda^b$) dynamicity computed in dynamic networks comprised of $T$ SINs.

| Metric | Equation | Description |
|---|---|---|
| $sim_1(a,b)$ | $min(\sum_{l=1}^{L} d(\delta_{nl}^a, \delta_{ml}^b))$ | Temporal similarity of structural dynamicity measured using Dynamic Time Warping (DTW) Technique |
| $sim_2(a,b)$ | $min(\sum_{l=1}^{L} d(\delta_{nl}^a, \delta_{ml}^b))$ | Temporal similarity of neighbourhood dynamicity measured using Dynamic Time Warping (DTW) Technique |
| $sim_3(a,b)$ | $\dfrac{\sum_t [(\delta^a(t) - \overline{\delta^a})(\delta^b(t) - \overline{\delta^b})]}{\sqrt{\sum_t (\delta^a(t) - \overline{\delta^a})^2 \sum_t (\delta^b(t) - \overline{\delta^b})^2}}$ | Correlation between structural dynamicity of two non-connected actors computed using Pearson correlation |
| $sim_4(a,b)$ | $\dfrac{\sum_t [(\lambda^a(t) - \overline{\lambda^a})(\lambda^b(t) - \overline{\lambda^b})]}{\sqrt{\sum_t (\lambda^a(t) - \overline{\lambda^a})^2 \sum_t (\lambda^b(t) - \overline{\lambda^b})^2}}$ | Correlation between neighbourhood dynamicity of two non-connected actors computed using Pearson correlation |
| $sim_5(a,b)$ | $1 - \dfrac{\sum_{t=1}^{T} [|\delta^a(t) - \delta^b(t)| + |\lambda^a(t) - \lambda^b(t)|]}{\sum_{t=1}^{T} [|\delta^a(t) + \delta^b(t)| + |\lambda^a(t) + \lambda^b(t)|]}$ | Similarity by the abundance of structural and neighbourhood dynamicity between two non-connected actors computed using Bray–Curtis dissimilarity measure |

### 3.1.1. Undirected Networks

The first undirected network dataset comes from a reality mining project at the Massachusetts Institute of Technology (MIT) in 2004, where the actors were tracked with the help of their personal smartphones to study interpersonal interaction. It contains human contact and interaction data among 100 students over nine months. In this undirected network, an actor in the network represents a person, and a link indicates physical contact between two persons. The second dataset comes from internal email communications among employees of a mid-sized manufacturing company where actors represent employees and links represent individual emails between two employees. This dataset was collected from Network Repository as an undirected dynamic network. The next dataset in this category contains undirected network data from a Facebook-like social network originating from an online community for students at the University of California, Irvine, where actors represent students within the community and links represent messages communicated among them. The last undirected network dataset is a very small subset of the total 'Facebook' friendship graph where an actor represents a Facebook user, and a link represents a friendship between two users. For the sake of brevity, we name these four networks as $G_{MIT}$, $G_{Email}$, $G_{UCI}$, and $G_{FF}$ to denote the network originated from the MIT reality project, a small manufacturing company, University of California Irvine and real Facebook Friendship, respectively, in the rest of the study. In these network datasets, links are date stamped with individual dates, and the smallest temporal granularity of these networks is a day. Therefore, three different sliding window sizes (i.e., $\tau = 1, 7$ and 30 days) were considered for sampling the longitudinal networks. This will help this study to emulate daily, weekly and monthly dynamic networks.

### 3.1.2. Directed Networks

The first directed network is constructed from the retweeting functionality of Twitter. There are predominantly two types of dynamic social networks that can be developed using the Twitter social network platform by considering its retweeting and mentioning mechanism. The first considers the fact that a Twitter user is reposting another user's tweet, and the latter considers one user mentioning the other in his/her tweet by using *@username*. Both types of networks built upon the Twitter information diffusion mechanism are directed networks. The Twitter network dataset in this study, collected from Network Repository, is a retweet network. Actors in this network are Twitter users, and a link

between them denotes whether the users retweeted each other. Since all links in the retweet network are time-stamped, this study used three different sliding window sizes (i.e., $\tau = 6$, 12 and 24 h) to sample the longitudinal networks considering the temporal granularity of hours. For the sake of brevity, this study used $G_{retwt}$ to denote the retweet network in the rest of the study.

### 3.1.3. Co-Authorship Networks

This study also considered two collaboration networks of authors of scientific papers in arXiv. Two sections were considered to build dynamic networks, namely, (i) high energy physics–phenomenology (Hep-ph) and (ii) high energy physics–theory (Hep-th). In these networks, a link between two authors represents a common publication that both authors have co-authored. $G_{ph}$ and $G_{th}$ are two symbols used to denote these two networks. Since the temporal granularity of these networks is a year, this study sampled dynamic networks considering yearly duration (i.e., $\tau = 365$ days) as the window size. In $G_{ph}$, yearly networks for the duration 1992–1998 were analysed in the training phase to predict the links in the year 1999 and, likewise, in $G_{th}$, the training phase was the duration 1993–1998, and the test phase was the year 1999. Table 2 sets out the basic and different temporal statistics for each type of network dataset. In this table, we explicitly describe the training and test duration for each dataset where the link prediction mechanism explores the link structure to predict the possibility of links during the test phase. In this study's context, the network within the training interval is split into smaller network snapshots known as short-interval networks (SIN) using three different sampling window sizes for each network but the co-authorship networks. In the latter case, we only used yearly window size. In the case of the undirected networks, we emulate daily, weekly and monthly networks by considering one-, seven-, and thirty-day sliding window sizes to sample dynamic networks and generate temporal network snapshots. On the other hand, in the case of the directed retweet network, we consider the hourly window as a sampling granularity. In the table, we observe that the cut-off time for this directed network is 4:00 a.m. Considering the timestamps, we selected 6, 12 and 24 h sampling window sizes to split this network and generate SINs. Considering this different sampling window size to aggregate links, each network dataset had different numbers of SINs in each dataset.

### 3.2. Supervised Link Prediction

The primary objective of the link prediction mechanism is to analyse the intrinsic characteristics of the network in regard to topological information or attributes related to actors or links among them in the training phase $[t_1, t']$ to predict the likelihood of future links in the test phase $[t', t_1']$. Since the purpose of this study is to predict links in dynamic networks in longitudinal settings, we therefore split the network $G_T[t_1, t']$ in the training phase into smaller temporal subnetworks considering different sliding window sizes to generate a time series or evolutionary sequence of network snapshots or SIN. Depending on the number of SINs in each dataset, the objective of this study is to build time series of two different dynamicity measures incident to actor pairs and generate dynamic features by considering the temporal similarity and correlation measures between two temporal sequences for the purpose of supervised link prediction. Supervised methods for link prediction problems need to predict possible future links by successfully discriminating between the links with positive and negative labels within a classification dataset. Thus, supervised link prediction turns into a binary classification task that involves learning positive and negative labels by exploiting interesting features describing each instance. Supervised link prediction setup starts with building classification datasets consisting of positive and negative instances. In this study's context, instances are the non-connected actor pairs from the network in the test phase $G_{T+1}[t_1, t']$. Instances are labelled as positive depending on their true appearances during the test phase, and links with negative labels were randomly selected from links that do not appear both in the training and test phases. This study considered a workload ratio of links with positive and

negative labels to 1:10. Thus, the number of negatively labelled links is ten times higher than those with positive labels in each classification dataset. However, in the case of the co-authorship networks, the workload ratio is 1:2. For the sake of simplicity in the link prediction problem, loops (i.e., links where source and destination are the same actors) were ignored and links that are unique in $G_{T+1}$ are considered (i.e., links not present in $G_T$. Choosing the appropriate feature set to describe instances in the classification dataset is one of the most important tasks in supervised link prediction. In each classification dataset of this study, both positively and negatively labelled actor–pair instances were described using features $sim_i(v_i, v_j)$ (i.e., metrics summarized in Table 1) those were developed considering the similarity of the evolutionary information associated with each actor of a pair. This study constructed different classification datasets consisting of instances and dynamic features describing those instances. Depending on the number of sampling window sizes, for both directed and undirected networks, each network dataset had three classification datasets and altogether 15 classification datasets. In the case of the co-authorship network datasets, since there was only one sampling window size (i.e., one year), both datasets had one classification dataset defined for this study's purpose.

**Table 2.** Basic statistics of network datasets used in this study. The training duration represents the interval used to generate temporal short-interval networks and the sampling interval denotes the sliding window sizes used to sample dynamic networks. SINs represent the number of short-interval networks or network snapshots generated using the corresponding window size.

| Dataset | Actors | Links | Training Duration | | Testing Duration | | Sampling Interval | SINs |
|---|---|---|---|---|---|---|---|---|
| | | | **Start** | **End** | **Start** | **End** | $\tau$ | |
| $G_{MIT}$ | 96 | 1,086,404 | 14 September 2004 | 31 January 2005 | 1 February 2005 | 5 May 2005 | 1 day | 140 |
| | | | | | | | 7 days | 20 |
| | | | | | | | 30 days | 5 |
| $G_{Email}$ | 167 | 82,927 | 2 January 2010 | 31 July 2010 | 1 August 2010 | 30 September 2010 | 1 day | 186 |
| | | | | | | | 7 days | 31 |
| | | | | | | | 30 days | 8 |
| $G_{UCI}$ | 1899 | 61,734 | 24 March 2004 | 31 May 2004 | 1 June 2004 | 26 October 2004 | 1 day | 45 |
| | | | | | | | 7 days | 7 |
| | | | | | | | 30 days | 3 |
| $G_{FF}$ | 11,715 | 42,698 | 1 January 2007 | 31 March 2007 | 1 April 2007 | 30 April 2007 | 1 day | 90 |
| | | | | | | | 7 days | 13 |
| | | | | | | | 30 days | 3 |
| $G_{retwt}$ | 14,370 | 39,124 | 14 September 2010 4 a.m. | 14 October 2010 4 a.m. | 14 October 2010 4 a.m. | 15 October 2010 4 a.m. | 6 h | 121 |
| | | | | | | | 12 h | 61 |
| | | | | | | | 24 h | 31 |
| $G_{th}$ | 6798 | 290,597 | 1 October 1993 | 31 December 1998 | 1 January 1999 | 10 December 1999 | 1 year | 6 |
| $G_{ph}$ | 16,959 | 2,322,259 | 15 March 1992 | 31 December 1998 | 1 January 1999 | 31 December 1999 | 1 year | 7 |

### 3.3. Performance Evaluation

As mentioned earlier, this study utilized dynamic features, computing similarity or correlation between two actors by considering network dynamics and using their evolutionary information, as described in Section 2, to describe both positively and negatively labeled instances in the classification datasets. Dynamic feature values were normalized such that the distribution has zero mean and one standard deviation. In regard to classifiers, this study used simple logistic regression, Random Forest, and Bagging algorithms. The latter two algorithms use ensemble-based methods. The well-known machine learning library WEKA [53] was used for classification purposes using the default parameters. For validation purposes, this study considered 10-fold cross-validation, and the mean scores

were used to determine the accuracy of the results. In addition to accuracy measures, AU-CROC (Area Under Receiver Operating Characteristics Curve) and AUCPR (Area Under Precision-Recall Curve) were also used to measure the classification performance. While the AUCROC measure is the de facto standard for measuring supervised learning-based classification, AUCPR is reported for a more differentiated view regarding the learning task in the imbalanced dataset. Despite its criticism [54], AUCROC is a popular metric (after accuracy) used in binary classification. Accuracy only classifies the class label as right or wrong; however, AUCROC quantifies the uncertainty associated with classifiers by introducing a probability value. It is essentially equivalent to average precision, which is another common measure for ranking results (Manning et al. 2008) [55] , and it relates the true positive rate against the true negative rate of a classifier. As an important traditional measure, it is also used in imbalanced classification problems. AUCROC score interprets the probability that a randomly chosen missing link in the test phase belongs to $G_{T+1}$ is given higher probability score than a randomly chosen non-existent link, which is absent both in the training $G_T$ and test network $G_{T+1}$. The calculation of AUCROC is given as defined by the formula AUCROC= $\frac{(n'' + 0.5n'')}{n}$, where $n$ denotes the number of independent comparisons, $n'$ denotes the times where a missing link in the test network has been given a higher score, and $n''$ denotes the times when a non-existent link has been given a higher score. AUCROC curve demonstrates how the number of correctly classified positive examples varies with the number of incorrectly classified negative examples and shows an overly optimistic view of an algorithm's performance in the presence of large skew in the class distribution where the precision–recall curve was proposed as an alternative to ROC in such cases. This is because a large change in the number of false positives can make a small change in the false positive rate (i.e., $\frac{FP}{FP+TN}$) that is used in constructing the ROC curve. Precision compares false positives to true positives instead of true negatives and captures the effect of many negative examples impacting the algorithm's performance. Furthermore, for a given ROC curve, the corresponding P–R (precision–recall) curve varies with skewness in the class distribution. Therefore, Boyd et al. [56] recommended that the area under the precision–recall curve (AUCPR) often serves as a summary statistic while comparing the performances of several different algorithms. The authors also proposed a method to determine the minimum value of AUCPR as $AUCPR_{min} = 1 + \frac{(1-\pi)ln(1-\pi)}{\pi}$ with skew $\pi = \frac{\#positivesamples}{n}$, where $n$ = total number of samples in the classification dataset. According to this equation, considering the ratio of positive and negative samples as 1:10 (i.e., the ratio of positive and negative samples is 1:10 in this study) in the classification datasets of the directed and undirected networks and the value of the skew $\pi = 0.091$, the minimum value of AUCPR in these datasets should be 0.04. However, for the rest of the three networks (i.e., $G_{ph}$ and $G_{th}$), since the skew $\pi = 0.33$ (i.e., ratio 1:2), the minimum value of AUCPR should be 0.189. For sake of comparison, this study compared the performances of dynamic features with a well-known metric, 'ResourceAllocation' [57], which is widely used for link prediction purposes in static networks and demonstrated improved performance. We also implemented the link prediction strategy in dynamic networks proposed by Soares and Prudêncio [34], where the authors built a time series of traditional topological metrics (e.g., Commonneighbours) for non-connected actor pairs for each SIN in the training phase. The authors also used the time series forecasting method (e.g., ARIMA) to predict the final score of the topological metrics and used those forecasted values to train the classifier. Different variations of this method are also extensively followed by other authors to support link prediction in dynamic networks [36,58]. For the sake of brevity, in the rest of the study, we used $sim_{RA}$ and $sim_{Soares}$ to denote the values computed for the positively and negatively labeled actor pairs considering *ResourceAllocation* metric and dynamic link prediction strategy proposed by Soares and Prudêncio [34]. It is noteworthy that, to compute $sim_{Soares}$, we have considered the well-known *Jaccard Coefficient* measure as the topological similarity metric and used the ARIMA forecasting method to predict the future values of the common neighbours incident to actor pairs.

## 4. Results

In this section, we describe the performance of the constructed dynamic similarity metrics in a supervised link prediction setup.

### 4.1. Classification Performance

Table 3 sets out the performance scores of three different classifiers in classifying positive and negatively labeled links. Classifier performances are demonstrated considering three different performance metrics described before. In regard to the accuracy score, this study observes that both linear (i.e., logistic regression) and ensemble-based classifiers perform reasonably well using the dynamic similarity metrics/dynamic features constructed in this study. Nevertheless, RandomForest, the ensemble-based classifier, outperformed others in the case of undirected and co-authorship networks, whereas the linear classifier performs well in the directed network. In regard to the ensemble-based classifiers, bagging with a decision tree used as a base classifier is susceptible to overfitting and computationally expensive, as it considers all the available features to split a node in decision trees. Conversely, the RandomForest, a special case of bagging, randomly considers only a subset of the best features of those available. Therefore, it performed superior to bagging in some cases. In the co-authorship networks, we observed better performance by the bagging algorithm considering all three performance metrics. Considering AUCROC scores in the RandomForest classifier across the classification datasets, this study observed better performance in $G_{MIT}$, and in the rest of the network datasets, scores are better in comparison to a random algorithm having the highest AUCROC score of 0.50. The highest AUCROC scorer is the RandomForest algorithm in $G_{UCI}$, whereas, in co-authorship networks, the highest AUCROC score was achieved in $G_{ph}$ by the same classifier. In regard to the AUCPR scores, in all networks, we have observed that all classifiers surpassed the minimum score of 0.04. Although the RandomForest algorithm demonstrated improved performance in most cases using this metric, interestingly, it demonstrated inferior performance in regard to AUCPR in $G_{ph}$ dataset. The highest AUCPR score achiever is the Bagging algorithm in $G_{th}$. Considering the aforementioned discussion on classifiers' performances, we found that the RandomForest algorithm in the undirected networks, the linear classifier, logistic regression in the directed network, and Bagging in the co-authorship networks exceeded others in regard to performance metrics. Although a further study can reveal the underlying reason behind the classification performance differences demonstrated by different classifiers, it is evident from the classifiers' performances in this table, and we can conclude that the dynamic similarity metrics constructed in this study can be successfully utilized to predict future links in dynamic networks. Considering the Random Forest classifier, in $G_{MIT}$, considering accuracy and AUCROC scores, although better performance was observed in the classification dataset constructed using a monthly sliding window (i.e., 30 days); however, considering the AUCPR score, it is the window size of seven days that performs well. Likewise, the similar window size provided better performance in $G_{Email}$ and $G_{FF}$. However, in $G_{UCI}$, we observed better performance in the classification datasets where the dynamic features were constructed using a window size of one day. On the other hand, in the directed network $G_{Retwt}$, considering the linear classifier, we have observed that the classification dataset, built upon considering hourly window size, has outperformed others. Based on the aforementioned observations, it is evident that the choice of the window resolution to sample or aggregate links in a dynamic network greatly impacts dynamic link prediction considering similarity/proximity metrics built upon the evolutionary information. This study took the advantage of three different algorithms provided in the WEKA machine learning software. In Table 4, we provide a comparable picture of these features in regard to their rank of importance obtained by these algorithms. It is noteworthy that, for each network dataset, we considered the high-performing classification datasets. These include the classification datasets in $G_{MIT}$, $G_{Email}$, and $G_{FF}$ using monthly window size, daily window in $G_{UCI}$ and hourly window size for $G_{Retwt}$. The ranks of the features are assigned in increasing order, with one denoting the highest ranking. Information gain and chi-square

evaluator algorithms evaluate the worthiness of a feature by calculating the information gain and chi-square statistics with respect to the class variable. On the other hand, the last two columns denote the rank of a feature in regard to the SVM and RandomForest classifier. Finally, all the ranks for every four algorithms were aggregated to generate the final rank. From this table, it is observable that $sim_5(a, b)$, which represents the Bray–Curtis similarity of actor dynamicity values, became the most prominent feature in $G_{MIT}$, and temporal similarity of neighbourhood dynamicity measured by the DTW method and denoted by $sim_2(a, b)$ was found as an important feature in $G_{Email}$, $G_{UCI}$, $G_{Retwt}$, $G_{FF}$ and $G_{ph}$. On the other hand, the temporal similarity of structural dynamicity values of actor pairs, denoted by $sim_1(a, b)$ turned out to be a leading feature in $G_{th}$. Features generated by considering the correlation of dynamicity values became the least significant features in most datasets.

### 4.2. Feature Importance

By considering the improved performances of our dynamic features, our next objective is to compare all dynamic features to assess their relative competency in the dynamic link prediction task. For this purpose, an alternative to ROC curves for models with a large skew in the class distribution. P–R curves can sometimes expose differences between classifiers that are not apparent in the ROC curves. In Figure 3, we present the ROC and P–R curves for the best-performing classification dataset from each group. For this figure, we selected a classification dataset from $G_{UCI}$ constructed using a daily temporal sampling window, $G_{Rtwt}$ with an hourly sampling window, and $G_{th}$ from the co-authorship networks. In regard to the classifier, we have selected RandomForest for the undirected network, logistic regression for the directed network, and bagging for the directed network. In this figure, the left column contains the P–R curves, and the right column presents the ROC curves in the three datasets chosen above. In this figure, the red line denotes the best-performing dynamic feature in the chosen classification dataset of each network, the green line denotes an existing time series of topological similarity metrics-based link prediction strategy, and the orange line denotes the performance of *ResourceAllocation* metric used in link prediction over the static network. The red lines in the P–R and ROC plots denote the high-performing dynamic feature in the corresponding datasets (i.e., $sim_2(a, b)$ in $G_{UCI}$ and $G_{Rtwt}$ and $sim_1(a, b)$ in $G_{th}$. The green lines denote the metric (i.e., $sim_{Soares}$) developed by following the method described by Soares and Prudêncio [34]. In this dynamic link prediction strategy, this study first developed a time series of the Jaccard similarity measure for every non-connected actor pair for each SIN generated, considering the sampling window size of the high-performing classification dataset of the corresponding network. For example, the high-performing classification dataset in $G_{UCI}$ was developed using daily SINs (see Table 3 for performance measures). Then, the ARIMA forecasting method was used to predict the future value of the Jaccard metric, which was also used to train the classifier for positive and negatively labeled dyads. Finally, the orange lines denote the traditional topological similarity metric ResourceAllocation (i.e., $sim_{RA}$) that is widely used for higher performance in predicting links in cross-sectional networks. It is noteworthy that, in P–R plots, the curves tend to lie in the bottom left corner of the diagonal line, whereas, in ROC plots, the curves tend to lie in the top-left region of the plots. The higher the curves than the diagonal line, the higher the predictor's performance. Through the curves, it is observable that, apart from the P–R curve in the $G_{Rtwt}$ network, in every plot, the dynamic features constructed in this study exceeded others in performance.

**Table 3.** Classification performances by three classifiers considering the classification datasets of undirected, directed networks and co-authorship networks considering three different window sizes used to sample dynamic networks. Both directed and undirected network datasets used three different sampling window sizes to generate SINs in the dynamic networks, whereas the co-authorship networks used only a yearly sliding window.

| | **Undirected Network** | | | | | | | | |
|---|---|---|---|---|---|---|---|---|---|
| | RandomForest | | | | | | | | |
| | Accuracy (%) | | | AUCROC | | | AUCPR | | |
| Days | 1 | 7 | 30 | 1 | 7 | 30 | 1 | 7 | 30 |
| $G_{MIT}$ | 82.19 | 80.52 | 84.91 | 0.683 | 0.663 | 0.700 | 0.30 | 0.46 | 0.29 |
| $G_{Email}$ | 76.29 | 87.47 | 88.23 | 0.714 | 0.644 | 0.724 | 0.40 | 0.32 | 0.31 |
| $G_{UCI}$ | 89.46 | 84.95 | 84.67 | 0.764 | 0.713 | 0.654 | 0.34 | 0.29 | 0.29 |
| $G_{FF}$ | 85.03 | 84.98 | 85.33 | 0.687 | 0.636 | 0.773 | 0.39 | 0.36 | 0.43 |
| | Bagging | | | | | | | | |
| $G_{MIT}$ | 70.69 | 71.71 | 71.77 | 0.611 | 0.614 | 0.671 | 0.33 | 0.44 | 0.31 |
| $G_{Email}$ | 77.22 | 77.69 | 75.96 | 0.656 | 0.594 | 0.603 | 0.34 | 0.27 | 0.33 |
| $G_{UCI}$ | 84.47 | 83.81 | 82.99 | 0.630 | 0.619 | 0.632 | 0.29 | 0.31 | 0.28 |
| $G_{FF}$ | 73.11 | 72.80 | 72.22 | 0.622 | 0.588 | 0.644 | 0.35 | 0.32 | 0.39 |
| | Logistic Regression | | | | | | | | |
| $G_{MIT}$ | 73.30 | 72.22 | 72.68 | 0.536 | 0.613 | 0.590 | 0.26 | 0.38 | 0.22 |
| $G_{Email}$ | 78.23 | 77.91 | 78.13 | 0.654 | 0.637 | 0.563 | 0.36 | 0.30 | 0.25 |
| $G_{UCI}$ | 85.25 | 84.73 | 84.64 | 0.628 | 0.573 | 0.619 | 0.29 | 0.26 | 0.22 |
| $G_{FF}$ | 75.44 | 75.22 | 75.03 | 0.664 | 0.618 | 0.579 | 0.40 | 0.35 | 0.27 |
| | **Directed Network** | | | | | | | | |
| | RandomForest | | | | | | | | |
| Hours | 6 | 12 | 24 | 6 | 12 | 24 | 6 | 12 | 24 |
| $G_{Retwt}$ | 87.87 | 87.59 | 87.03 | 0.739 | 0.712 | 0.720 | 0.36 | 0.26 | 0.26 |
| | Bagging | | | | | | | | |
| $G_{Retwt}$ | 85.81 | 84.55 | 85.11 | 0.695 | 0.644 | 0.574 | 0.21 | 0.21 | 0.19 |
| | Logistic Regression | | | | | | | | |
| $G_{Retwt}$ | 88.11 | 88.13 | 88.01 | 0.735 | 0.712 | 0.622 | 0.32 | 0.26 | 0.23 |
| | **Co-Authorship Network (Window Size = 1 Year)** | | | | | | | | |
| | RandomForest | | | | | | | | |
| | Accuracy | | | AUCROC | | | AUCPR | | |
| $G_{th}$ | 77.90 | | | 0.663 | | | 0.49 | | |
| $G_{ph}$ | 81.49 | | | 0.722 | | | 0.18 | | |
| | Bagging | | | | | | | | |
| $G_{th}$ | 81.35 | | | 0.702 | | | 0.56 | | |
| $G_{ph}$ | 80.95 | | | 0.711 | | | 0.32 | | |
| | Logistic Regression | | | | | | | | |
| $G_{th}$ | 66.45 | | | 0.593 | | | 0.43 | | |
| $G_{ph}$ | 70.90 | | | 0.581 | | | 0.11 | | |

**Table 4.** The rank of different dynamic features constructed in this study using different algorithms for directed, undirected and co-authorship networks. Ranks are in increasing order with number one denoting the highest ranking. The total column represents the aggregation of all ranking scores to generate the final ranking.

| Feature Name | Information Gain | Chi-Square Attribute Evaluation | Support Vector Machine Evaluator | Random Forest Evaluator | Total |
|---|---|---|---|---|---|
| | | | $G_{MIT}$ | | |
| $sim_1(a,b)$ | 5 | 5 | 1 | 4 | 15 |
| $sim_2(a,b)$ | 2 | 2 | 5 | 1 | 10 |
| $sim_3(a,b)$ | 3 | 3 | 3 | 5 | 14 |
| $sim_4(a,b)$ | 4 | 4 | 4 | 3 | 15 |
| $sim_5(a,b)$ | 1 | 1 | 2 | 2 | 6 |
| | | | $G_{Email}$ | | |
| $sim_1(a,b)$ | 5 | 5 | 3 | 3 | 16 |
| $sim_2(a,b)$ | 2 | 2 | 1 | 2 | 7 |
| $sim_3(a,b)$ | 3 | 3 | 5 | 1 | 12 |
| $sim_4(a,b)$ | 4 | 4 | 2 | 4 | 14 |
| $sim_5(a,b)$ | 1 | 1 | 4 | 5 | 11 |
| | | | $G_{UCI}$ | | |
| $sim_1(a,b)$ | 2 | 2 | 1 | 2 | 7 |
| $sim_2(a,b)$ | 1 | 1 | 2 | 1 | 5 |
| $sim_3(a,b)$ | 5 | 5 | 3 | 4 | 17 |
| $sim_4(a,b)$ | 4 | 4 | 5 | 5 | 18 |
| $sim_5(a,b)$ | 3 | 3 | 4 | 3 | 13 |
| | | | $G_{FF}$ | | |
| $sim_1(a,b)$ | 3 | 3 | 4 | 3 | 13 |
| $sim_2(a,b)$ | 1 | 1 | 3 | 1 | 6 |
| $sim_3(a,b)$ | 5 | 5 | 5 | 5 | 20 |
| $sim_4(a,b)$ | 2 | 2 | 1 | 2 | 7 |
| $sim_5(a,b)$ | 4 | 4 | 2 | 4 | 14 |
| | | | $G_{Rtwt}$ | | |
| $sim_1(a,b)$ | 2 | 2 | 1 | 2 | 7 |
| $sim_2(a,b)$ | 1 | 1 | 2 | 1 | 5 |
| $sim_3(a,b)$ | 5 | 5 | 5 | 3 | 18 |
| $sim_4(a,b)$ | 4 | 4 | 4 | 5 | 17 |
| $sim_5(a,b)$ | 3 | 3 | 3 | 4 | 13 |
| | | | $G_{th}$ | | |
| $sim_1(a,b)$ | 2 | 2 | 3 | 1 | 8 |
| $sim_2(a,b)$ | 1 | 1 | 5 | 3 | 10 |
| $sim_3(a,b)$ | 5 | 5 | 1 | 5 | 16 |
| $sim_4(a,b)$ | 3 | 3 | 2 | 2 | 13 |
| $sim_5(a,b)$ | 4 | 4 | 4 | 4 | 16 |
| | | | $G_{ph}$ | | |
| $sim_1(a,b)$ | 5 | 5 | 2 | 4 | 16 |
| $sim_2(a,b)$ | 3 | 3 | 1 | 1 | 8 |
| $sim_3(a,b)$ | 2 | 2 | 3 | 2 | 9 |
| $sim_4(a,b)$ | 1 | 1 | 5 | 5 | 12 |
| $sim_5(a,b)$ | 4 | 4 | 4 | 3 | 15 |

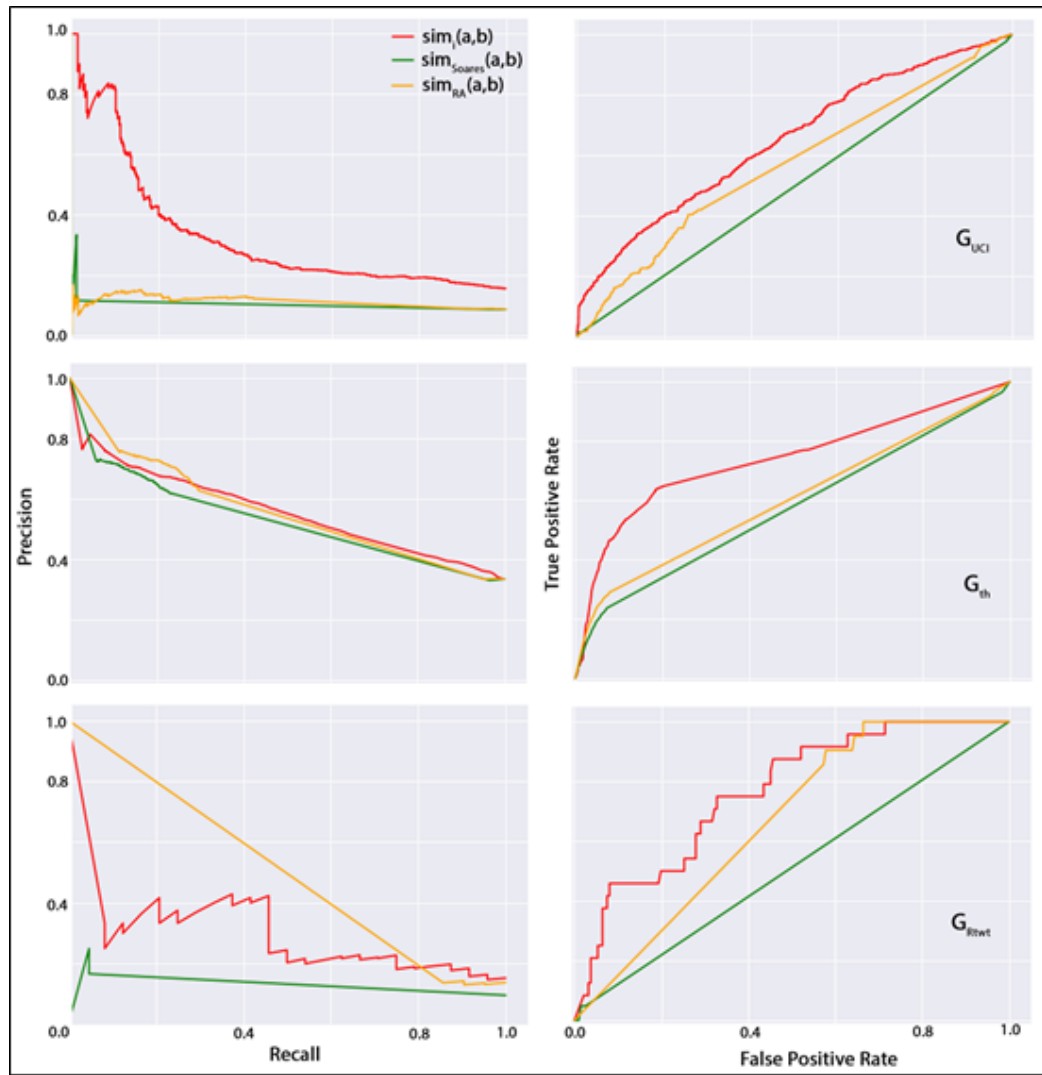

**Figure 3.** P–R curves (**left column**) and ROC curves (**right column**) for undirected network $G_{UCI}$ (**top row**), co-authorship network $G_{th}$ (**middle row**), and directed network $G_{Rtwt}$ (**bottom row**).

## 5. Discussion and Conclusions

Link prediction problems in social networks gained considerable research interest from various domains including anthropology, sociology, biology, information and computer science. Focusing on static network topological information without considering the influence of evolutionary process and associated dynamic changes incident to all actors in the temporal network, has led the existing methods to be incompetent in dynamic link prediction despite their compliance with the performance expectations. Recently, scholars tend to acknowledge that emerging links can be derived by mining the evolutionary information extracted from the network snapshots over time. Dynamic network topology along with associated evolutionary information resulting from the temporal, structural, and neighbourhood changes, associated with individual actors, can be exploited in dynamic link prediction. Furthermore, since most networks inherently evolve over time, it is imperative to exploit the temporal network dynamics to resolve issues with link prediction problems in dynamic networks.

Considering the problem of dynamic link prediction, this study attempted to develop evolutionary features by measuring the temporal similarity and correlation of the actor-oriented evolutionary information in dynamic networks. For this purpose, we considered both directed and undirected social networks of different sizes and domains. In a dynamic network composition, each selected network was sampled by considering three

different resolutions ranging from hours and days to years to generate SINs. Considering different temporal granularity, this study then developed a time series of structural and neighbourhood changes experienced by each user. Considering the rate of changes, this study then defined two temporal measures to quantify actors' temporal behaviour. These measures are defined as structural and neighbourhood dynamicity. To develop the dynamic features, this study leveraged the evolution of temporal similarity, and the correlation of these two dynamicity values to quantify the similarity/proximity between actors from an evolutionary perspective. The first four dynamic similarity metrics were constructed in this way. The fifth dynamic feature was constructed by considering the similarity measures widely used in ecology. In this measure, we quantify the normalised abundance of actor-level dynamicity in temporal networks. In a supervised link prediction setup, we have used two ensemble-based classifiers and one linear classifier to measure the performance of our dynamic features. Through the performance metrics used in this study, we have observed that these features can not only be indulged for dynamic link prediction purposes but also can effectively support modelling the network growth. The performance of dynamic features was also compared with a traditional topological metric widely used for link prediction purposes in cross-sectional networks and a time series-based dynamic link prediction strategy. In both cases, we have observed that dynamic features constructed by leveraging the evolutionary aspect of actors can perform as well as the traditional ones and sometimes can outweigh them in regard to prediction performance.

This study can further be extended in different ways. For example, instead of considering the network structural changes, one can exploit the temporal clustering tendency of actors, introduce time series forecasting methods to predict the future values of actor-level changes and employ other similarity measures (e.g., Euclidean, Manhattan) to measure the similarity between temporal information. One important observation we have noticed in this study is that the choice of the optimal sliding window to sample dynamic networks can effectively impact prediction performance. Therefore, further study can exploit dynamic link prediction performance to determine the optimality of the sampling resolution. On the other hand, in the case of the directed networks, this study used the aggregation of in-degree and out-degree centrality and neighbours to define the structural and neighbourhood dynamicity. Further studies can adopt these centrality measures separately to observe the prediction performance variations. Finally, like many other applications of link prediction problems, this study can be valuable to help define new dynamic similarity metrics for the purpose of dynamic link predictions in networks that inherently evolve over time including terrorist networks, online social networks (e.g., Twitter), scholarly and knowledge networks (e.g., keyword network) and collaborative filtering for the purpose of modelling the consumers' buying behaviour.

**Author Contributions:** Conceptualization, N.C. and S.U.; methodology, N.C.and S.U.; investigation, N.C.; writing—original draft preparation, N.C.; writing—review and editing, N.C. All authors have read and agreed to the published version of the manuscript.

**Funding:** This research received no external funding.

**Institutional Review Board Statement:** Not applicable.

**Informed Consent Statement:** Not applicable.

**Data Availability Statement:** No new data were created or analyzed in this study. Data sharing is not applicable to this article.

**Conflicts of Interest:** The authors declare no conflict of interest.

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
