# Peer review of "Evolutionary Features for Dynamic Link Prediction in Social Networks"

_applsci, doi:10.3390/app13052913_

Round 1

Reviewer 1 Report

The core of this work is to solve the Dynamic Link Prediction. The authors built dynamic similarity metrics by considering temporal similarity and correlation between different actor-level evolutionary information of non-connected actor pairs. Although experiment results show the superiority of their method, I think there still needs to be some improvements before publishing in Applied Sciences.

1 The major drawback of the article is the lack of a state-of-the-art with a real comparison to other dynamic similarity metrics, instead, the different approaches are quickly described in the introduction.

2 You should also consider the reproducibility of your work and make the code of your method available and the code of the experiments.

3 There are many articles from five years ago. Authors should introduce some latest developments in this field.

4 Some grammatical and letter errors should be carefully corrected, for example, in line 673. Moreover, tables should be in three-wire tables, and formulas should be in MathType format.

5 You should also consider the reproducibility of your work, and make the code of your method available, as well as the code of the experiments.

Author Response

The core of this work is to solve the Dynamic Link Prediction. The authors built dynamic similarity metrics by considering temporal similarity and correlation between different actor-level evolutionary information of non-connected actor pairs. Although experiment results show the superiority of their method, I think there still needs to be some improvements before publishing in Applied Sciences.

Comment 1: The major drawback of the article is the lack of a state-of-the-art with a real comparison to other dynamic similarity metrics; instead, the different approaches are quickly described in the introduction.

Response: We thank the reviewer for this valuable comment. Our link prediction method in dynamic networks in this article developed dynamic features that are considered as proximity measures using topological properties for predicting the connection probability between a pair of non-connected actors. This problem has been addressed by using different approaches by the researchers. Some of these methods are described in the introduction section in this article. These methods not only differ in approaches but also in performances.  Since the approach in the current study resembles supervised learning strategy, we intended to select one of the other supervised learning approaches that also constructed proximity features using topological and structural properties. Therefore, we selected the study by Soares, Prudêncio, & Cavalcante (da Silva Soares & Prudêncio, 2012). Besides, we also used similar datasets the author used in this study for performance comparison.

Comment 2: You should also consider the reproducibility of your work and make the code of your method available and the code of the experiments.

Response: The codes for this study were written in two different languages (i.e., C# and Python). For the network package, we used the NetworkX (Hagberg & Conway, 2020). By taking the reviewer’s comment into consideration, we plan to provide the code to calculate the Structural and Neighbrhood Dynamicity.

Comment 3: There are many articles from five years ago. Authors should introduce some latest developments in this field.

Response: We appreciate the respected reviewer to point this out and consequently, we discarded many older articles from the references and included some recent articles on the same topic published in years ranging from 2018-2021.

Comment 4:  Some grammatical and letter errors should be carefully corrected, for example, in line 673. Moreover, tables should be in three-wire tables, and formulas should be in MathType format.

Response: We went through the whole article again and in the updated version of the article we took great care to avoid the grammatical errors and syntax error in equations.

Comment 5: You should also consider the reproducibility of your work, and make the code of your method available, as well as the code of the experiments.

Response: responded earlier

Reviewer 2 Report

Please see my comments in the following attached file. 

Reviewer 3 Report

  • A dynamic similarity metric is proposed that considers how similar and related different actor-level evolution information is over time for pairs of actors who don't know each other.

  • I could not find recent citations after 2017. It is recommended to go through the recent articles as well and include them in the literature survey.

  • The captions of the figures and tables are too lengthy. Please reduce captions to one line only and give details in the text.

  • Line 729: "Conversely, P-R curves are used as an"; please finish it here.

Author Response

A dynamic similarity metric is proposed that considers how similar and related different actor-level evolution information is over time for pairs of actors who don't know each other.

Comment 1: I could not find recent citations after 2017. It is recommended to go through the recent articles as well and include them in the literature survey.

Response: In the revised manuscript, we included a set of most recent articles in the introduction section as our literary survey.

Comment 2: The captions of the figures and tables are too lengthy. Please reduce captions to one line only and give details in the text.

Response: We reduced the length of the text associated with the Figures 1, 2, and 3 and moved the addition descriptions to in-text.  

Comment 3: Line 729: "Conversely, P-R curves are used as an"; please finish it here.

Response: The corresponding text was a typo from the latex commenting error. We discarded the text and replaced with appropriate alternative.